# Can Language Models Solve Olympiad Programming?

**Quan Shi**[*], **Michael Tang**[*], **Karthik Narasimhan, Shunyu Yao**
Princeton Language and Intelligence (PLI), Princeton University
{qbshi, mwtang}@princeton.edu

## Abstract

Computing olympiads contain some of the most challenging problems for humans, requiring complex algorithmic reasoning, puzzle solving, in addition to generating efficient code. However, it has been understudied as a domain to evaluate language models (LMs). In this paper, we introduce the USACO benchmark with 307 problems from the USA Computing Olympiad, along with high-quality unit tests, reference code, and official analyses for each problem. These resources enable us to construct and test a range of LM inference methods for competitive programming for the first time. We find GPT-4 only achieves a 8.7% pass@1 accuracy with zero-shot chain-of-thought prompting, and our best inference method improves it to 20.2% using a combination of self-reflection and retrieval over episodic knowledge. However, this is far from solving the benchmark. To better understand the remaining challenges, we design a novel human-in-the-loop study and surprisingly find that a small number of targeted hints enable GPT-4 to solve 13 out of 15 problems previously unsolvable by any model and method. Our benchmark, baseline methods, quantitative results, and qualitative analysis serve as an initial step toward LMs with grounded, creative, and algorithmic reasoning.

## 1   Introduction

Code generation has become an important domain to evaluate and deploy language models (LMs). However, with the scaling of LMs and the development of new inference methods (Wei et al., 2022; Shinn et al., 2023; Chen et al., 2023; Zhou et al., 2022), many popular coding benchmarks such as HumanEval (Chen et al., 2021) and MBPP (Austin et al., 2021) have been saturated with solve rates above 90%. To drive further progress, we need more challenging benchmarks that reveal limitations of existing models and inference methods, and provide actionable insights for improving LM's algorithmic reasoning.

Competitive programming is a natural fit for this pursuit, as it has been designed to rigorously evaluate the human ability to reason about complex scenarios and create novel algorithms. However, previous explorations of competitive programming lack exhaustive unit test suites, lack problem analyses, or lack enough problem diversity to comprehensively evaluate algorithmic reasoning (Li et al., 2022; Hendrycks et al., 2021; Jain et al., 2024).

We thus introduce **USACO**, a carefully crafted coding benchmark with 307 challenging problems from past USA Computing Olympiad (USACO) competitions. As shown in Figure 1, each problem describes a task to solve in a fictional scenario, along with some example tuples of inputs, outputs, and explanations. Solving these problems not only require a wide range of algorithmic, mathematical, and commonsense knowledge, but also grounded and creative reasoning: unlike previous program synthesis benchmarks, successful models must reason over ad hoc environments, creating novel algorithms tailored to each problem scenario. On USACO, even the best LM (GPT-4) only reaches a zero-shot pass@1 solve rate of 8.7% using zero-shot chain-of-thought prompting.

To study more advanced inference-time methods on competitive programming, for the first time, our benchmark also collects high-quality unit tests, reference code solutions, and

---

[*]Equal contribution. Code, data, examples: https://princeton-nlp.github.io/USACOBench/.

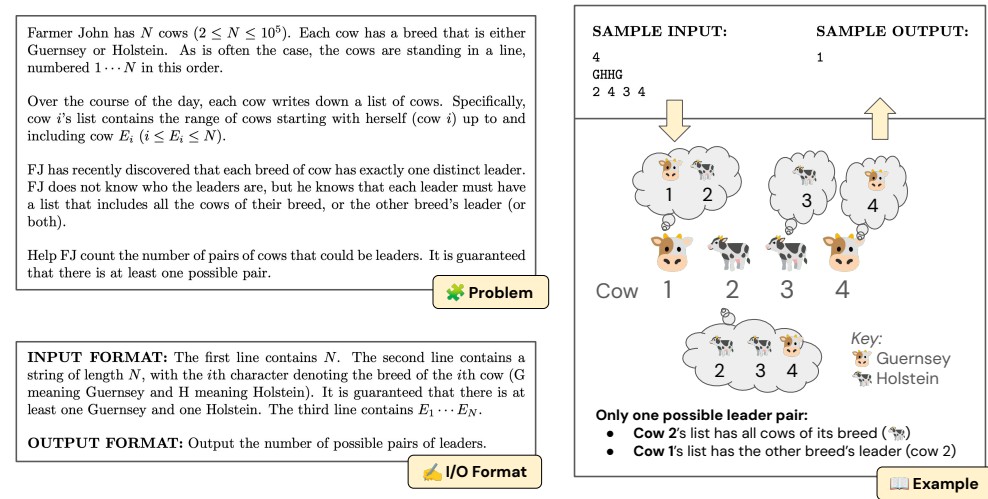

Figure 1: Example USACO problem description, formatting instructions, and illustration (problem id: `1275_bronze_leaders`). Solving this problem requires a combination of *grounded* reasoning about the concept of leaders, *creative* thinking to precisely count different cases of leader pairs, and *algorithmic* reasoning to perform these ad hoc operations in linear time.

official analysis for each problem, along with corresponding instructional texts in the form of competition programming textbooks. Based on these resources, we construct a range of baseline methods based on retrieval (Gao et al., 2023), self-reflection (Shinn et al., 2023; Chen et al., 2023), and their combinations. We find that combining retrieval over similar problems and solutions and self-reflection maximizes performance gains, well over **doubling** the zero-shot solve rate of GPT-4. However, all methods are still far from solving the benchmark above bronze level, the easiest difficulty tier.

To further understand the limitations and potentials of LM reasoning toward competitive programming, we perform a novel human study where humans interact with LMs in a conversational "tutoring" setup by pointing out errors and giving minimal hints. To our surprise, on a subset of 15 problems where GPT-3.5 and GPT-4 can never solve using any inference methods, such a human-in-the-loop setup leads to GPT-4 solving 13 out of 15 problems, whereas GPT-3.5 solves *none*. This indicates the emergent potential of stronger LMs to incorporate high-quality feedback, the need to develop new methods that can generate such human-level corrective feedback, and a re-thinking of the right metric for measuring model capabilities beyond the overly strict execution success.

To summarize, our contributions of our work are:

- We propose the USACO benchmark, the first benchmark based on olympiad programming with high quality test cases, problem analyses, and auxiliary resources.
- For the first time, we construct and test LM inference methods for Olympiad programming, such as self-reflection and retrieval. Our results indicate a combination of retrieval and self-reflection can significant boost performance, but is still far from solving the benchmark.
- We conduct a novel human-in-the-loop study to characterize the capabilities and limitations of LMs for Olympiad programming, complementary to automatic experiments based on execution success. We find that only certain models can successfully incorporate feedback, uncovering latent differences between models

## 2 Related Work

**Code Generation Benchmarks** Language model performance on simple program synthesis has been thoroughly explored (Yu et al., 2018; Chen et al., 2021; Austin et al., 2021; Zan

et al., 2022), with HumanEval being the general standard for benchmarking new models on code synthesis. However, current models, aided by inference techniques, can achieve up to a 94% solve rate on HumanEval (Zhou et al., 2023), indicating a need for harder, more complex, yet self-contained coding tasks to probe the upper limit of code reasoning. Competitive programming questions have thus been proposed as a more difficulty evaluation metric, with most problems coming from online platforms such as Codeforces, Atcoder, Kattis...(Li et al., 2023b; Huang et al., 2023; Jain et al., 2024; Hendrycks et al., 2021; Li et al., 2022). However, these problems largely do not contain quality problem analyses, comprehensive correctness-defining test cases, with many problems being presented purely symbolically. This means that it can only weakly evaluates the model's ability to creatively reason in grounded problem environments, a crucial ability of well-rounded reasoners.

**Inference Time Methods for LMs**   Inference time methods have seen significant success in improving reasoning capabilities by conditioning generations on environment feedback, task-specific knowledge, natural language reflections, and planned summaries (Shinn et al., 2023; Chen et al., 2023; Madaan et al., 2023; Yao et al., 2022; Zelikman et al., 2022; Zhou et al., 2023; Gao et al., 2023; Le et al., 2022). However, their utility on code domains has only previously been explored on simple program synthesis tasks such as HumanEval and MBPP (Chen et al., 2021; Austin et al., 2021). In this paper, we additional provide insights on their performance in competitive programming, a much more difficult domain. Our instantiation of retrieval augmented generation additionally takes inspiration from cognitive architectures for humans reasoning (Sumers et al., 2023) and classical case-based reasoning literature (Aamodt & Plaza, 1994; Schank, 1983), mirroring the types of information humans find useful for problem solving.

**Human Model Interaction**   Sumers et al. (2022) investigates agent learning from human provided feedback under synthetic tasks. Macina et al. (2023) aims to provide a tutoring ruleset to effectively engage LMs in dialogue math problem solving. In this paper we adopt a similar setup to code, applying a specified interaction ruleset to gauge the ability of models to respond to feedback.

## 3   The USACO Benchmark

The USACO benchmark consists of 307 high-quality expert-written problems from past USA Computing Olympiad contests (https://usaco.org). Each problem consists of a problem description with instructions for reading and writing from standard input and output; 0-2 sample tests; 10-17 hidden tests verifying solution correctness; time and memory limits verifying solution complexity; and an official human-written *problem analysis* explaining the solution in detail with corresponding Python code.

**Problem Difficulty**   Problems are divided into tiers of increasing difficulty consisting of bronze, silver, gold, and platinum. At all levels, solutions typically require ad hoc algorithmic reasoning and, unlike interview-level problems, rarely follow directly from well-known algorithms. Gold and platinum problems may additionally require knowledge of known algorithms and data structures, often using them in unorthodox ways.

**Task Formulation**   A model is given the problem description, including any available samples, and time and memory limits. The model must then produce a code solution, which is run by the judge and accepted if it produces the expected outputs on all hidden tests under the given limits, enforcing both correctness and the desired asymptotic efficiency. Note that the model cannot get access to hidden test inputs or outputs, but can receive information on how many tests a given solution has passed.

**Task Features**   We find several features of USACO that make it an effective LLM evaluator. Firstly, USACO problems contain detailed problem environments encouraging grounded, creative reasoning. Problem narratives are high quality, and avoid purely symbolic coding questions such as many presented on platforms like LeetCode or Kattis. Thus, problems

| Benchmark | Exhaustive Unit Tests | Expert-Written Problem Analyses | Non-Symbolic Environments |
|---|:---:|:---:|:---:|
| HumanEval (Chen et al., 2021) | ✓ | ✗ | ✗ |
| APPS (Hendrycks et al., 2021) | ✗ | ✗ | ✓ |
| CodeContests (Li et al., 2022) | ✗ | ✗ | ✓ |
| USACO (ours) | ✓ | ✓ | ✓ |

Table 1: The USACO benchmark features high-quality problem environments with grounded creative reasoning, complete hidden tests, and expert-written problem analyses with both gold solution reasoning and Python code.

focus on a model's ability to reason creatively and ground insights and algorithmic designs to the details of a given scenario. Furthermore, problems are tiered in difficulty (see Table 3), with bronze tier problems serving as an effective test of pure reasoning, requiring no formal knowledge of data structures and algorithms. At higher tiers, problems go beyond basic implementation of algorithms, instead requiring creative, grounded algorithm design specific to the desiderata and constraints of the given scenario. Finally, the USACO benchmark includes expert-written problem analyses containing both natural language and gold Python solutions to all problems, enabling development of rich inference-time techniques and nuanced evaluation beyond unit test execution.

| Difficulty | Core Skills Evaluated |
|:---:|:---:|
| Bronze | simulation, complete search, sorting, greedy |
| Silver | binary search, comparators, graphs, trees, floodfill, prefix sums, bitwise operators |
| Gold | dynamic programming, disjoint set union, spanning trees, Euler tour, combinatorics |
| Platinum | segment tree, range queries, binary jumping, sweep line, convex hull, flows |

Table 2: Core skills evaluated at each tier of USACO, from `https://usaco.guide/`.

**Construction: Problems** We collect 484 problems from `https://usaco.org` detailing materials on contests hosted between 2011 and 2023 using a custom HTML parser, and use regular expressions to extract wall clock time and memory constraints from problem descriptions. which are manually verified using solutions with correct and incorrect asymptotics.

**Construction: Problem analyses** To assist the development of rich inference-time methods and evaluations, we select the 307 problems out of 484 with full problem analyses. We parse an English-only analysis without code, as well as a ground truth standalone Python 3 code snippet. For the majority of problems where Python code is unavailable, we prompt GPT-4 to translate the code to Python 3 and validate that all code solutions pass hidden tests on the given constraints.

### 3.1 Baseline Results

We begin by evaluating zero-shot performance of models representing state-of-the-art coding performance as a baseline: this includes GPT-3.5 (gpt-3.5-turbo-1106), GPT-4 (gpt-4-1106-preview), Claude-3-sonnet, (claude-3-sonnet-20240229), CodeLlama2-Instruct-7B, and Deepseek-Coder-Instruct-7B (Roziere et al., 2023; Guo et al., 2024; OpenAI et al., 2023). This is summarized in Table 3. Unless otherwise indicated, models were prompted with chain of thought (Wei et al., 2022), refer to figure 6 for full prompts. Following previous work on competitive programming (Li et al., 2022; Hendrycks et al., 2021), we evaluate primarily based on the unbiased pass@k metric defined in (Chen et al., 2021).

**Solve rates near zero for gold difficulty and above.** We find that USACO presents a strong challenge to current generation models. Weaker models like GPT-3.5, CodeLlama,

| Statistics | USACO |
|---|---|
| Number of problems | 307 |
| Avg words per problem | 452.9 |
| Avg words per problem analysis | 1107.8 |
| Bronze problems | 123 |
| Silver problems | 100 |
| Gold problems | 63 |
| Platinum problems | 21 |

| Model | Pass@1 |
|---|---|
| CodeLlama (7B) | 0.19 |
| DeepSeek Coder (7B) | 1.04 |
| GPT-3.5 | 0.59 |
| Claude-3-Sonnet | 2.61 |
| **GPT-4** | **8.7** |
| Human Average | 35.83 |

Table 3: Statistics and zero-shot pass@1 performances on USACO. Human performance estimated through past contest performance.

and DeepSeeker cannot solve any problem above silver difficulty, while newer models like GPT-4 have near-zero pass rates for gold difficulty problems and above. For full per-difficulty solve rates, refer to Appendix B.3.

**For stronger models, most errors are algorithmic.** Other than CodeLlama, we find that no model errors are significantly due to compilation errors. We detail full results in Appendix B.2. This shows at the very least, models are effective in generating syntactically correct code, and indicates more nuanced issues in generations such as problem misunderstandings. We document some samples of problems and their errors in Appendix 7 and perform brief qualitative analysis.

**Problem release date impacts performance.** We test temporal effects by additionally evaluating models on a small selection of 36 problems released after training cutoff dates. We find that solve rate drops to 0 for all models. However, we do note that USACO questions are well known to increase in difficulty every year, making this likely an effect of difficulty increases, inclusion of reasoning in pre-training data, as well as small sample size. Despite this, the overall performance is still quite poor.

## 4 Inference Time Techniques for Better Reasoning

Past work has demonstrated that curated prompting and retrieval strategies can significantly improve performance on various tasks across natural language processing, multi-task QA, and embodied intelligence (Yao et al., 2022; Wang et al., 2023; Madaan et al., 2023). To investigate the effectiveness of such inference-time methods, we adapt self-reflection and retrieval techniques widely successful in other domains to USACO.

### 4.1 Self-Reflection

Self-reflection techniques aims to allow models to iteratively improve generations by conditioning future output on execution feedback of previous attempts. We primarily experiment with Reflexion (Shinn et al., 2023), a representative technique that additionally maintains an episodic buffer of past attempts to induce better reasoning in future trials.

**Setup** The model is first prompted to solve the problem, generating a code solution as well as an explanation of the code. For each iteration of reflection, the model is prompted to first reflect on what went wrong previously, then fix the previous code, given the the execution output of the previous solution, the previous solution itself, as well as the contents of a buffer of past solve attempts. After each iteration, the previous attempt as well as the execution results is added to the buffer. This loop iterates until a maximum number of debugging steps is reached. We set this hyperparameter i = 3 as we observe no empirical gains in solve rate past 3 rounds of debugging: see Appendix C.5.

### 4.2 Retrieval Augmented Generation

Retrieval augmented generation (RAG) has similarly proven to reduce hallucinations and improve reasoning capabilities in a variety of domains. (Gao et al., 2023; Su et al., 2024; Shypula et al., 2023). However, it has seen limited usage in coding domains as it is difficult

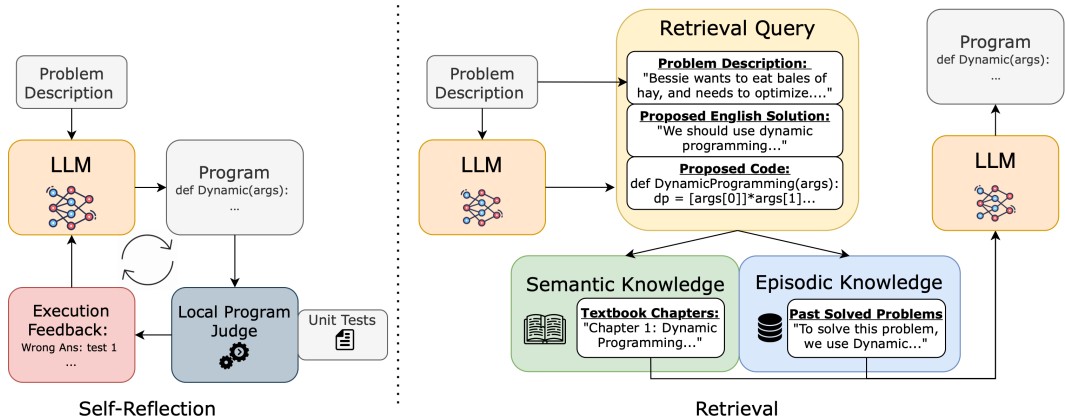

Figure 2: Overview of inference methods tested: self-reflection (left) prompts the model to review execution feedback to revise its generation; retrieval (right) uses the problem and a draft solution to query relevant semantic and episodic knowledge to generate a more informed final solution.

to pinpoint what types of knowledge is useful in aiding code generation. We note that when solving problems, humans tend to recall either task-specific, established algorithms, facts, and concepts about the domain, or past experiences and generalizations from solving previous, similar problems relative to the problem at hand. This represents semantic knowledge, and episodic knowledge respectively (Sumers et al., 2023). Inspired by this, we curate two setups we aptly name semantic retrieval and episodic retrieval, and instantiate the semantic knowledge store as a competitive programming textbook, and the episodic knowledge store as the bank of USACO problems and solutions not currently being solved.

**Semantic Knowledge Store**  We use the cp-algs textbook (`https://cp-algorithms.com`), which contains 30 human-written chapters on algorithmic concepts specifically targeted to the USA Computing Olympiad. Chapters contain English text as well as code snippets. Entire chapters barely fit within the context limit of GPT-4: therefore when lower context models (like GPT-3.5), or multiple types of knowledge are incorporated, we truncate the retrieved chapter to fit within the context length.

**Episodic Knowledge Store**  We simulate a setup where the model has seen all other problems in the USACO set except for the current one it is solving, simulating a k-fold evaluation that holds out one problem at a time: this allows us to maximally investigate the potential of retrieval on the relatively small dataset (we obtain similar results with a more traditional train-test split, as detailed in Appendix A. For each "seen" problem, its corresponding problem description, english solution, and python solution code is concatenated together to form documents to retrieve over. We tune over the number of problems to retrieve, $p$, and find that $p = 2$ is optimal for GPT-4, and $p = 1$ is optimal for GPT-3.5 and thus report these numbers. Full details can be found in Appendix C.5.

**Retrieval Query**  Ablations on retrieval queries indicate that the most effective retrieval query utilizes the current problem description, as well as an initial solution attempt containing code and english explanation. This allows accurate retrieve relevant algorithm descriptions from the underlying retrieval corpus, as solely utilizing the problem descriptions does not allow retrieval over algorithmic keywords. We do not count this initial generation as an attempt as it does not get evaluated by our local judge. Additional details on ablation experiments can be found in Appendix C.3.

**Setup**  Models are prompted with the problem description, and first generate an initial solution to be used in the retrieval query. This initial solution, along with the problem description, is fed into a BM25 retrieval function, and the highest ranking document is inserted into the context to aid the real solving process. Prompts can be found in Appendix C.7. We additionally note that retrieval-based methods and reflection methods are orthogonal and can be sequentially applied. Thus, we test settings combining various combinations of retrieval types, as well as retrieval combined with reflection.

| Technique | GPT-3.5 | GPT-4 |
|---|---|---|
| Zero-shot | 0.59 | 8.70 |
| Reflexion | 0.97 | 12.38 |
| Semantic Retrieval | 2.04 | 10.26 |
| Episodic Retrieval | 5.49 | 14.33 |
| Semantic + Episodic | 6.76 | 12.54 |
| Semantic + Reflexion | 2.64 | 11.82 |
| **Episodic + Reflexion** | **8.79** | **20.20** |
| Semantic + Episodic + Reflexion | 7.52 | 18.05 |

Figure 3: Pass@1 Performance of Reflexion and Retrieval based methods on USACO.

**Evaluation**    All methods are evaluated Pass@1.For self-reflection, we adapt methodology in Shinn et al. (2023), and provide only the execution results of the exposed sample test cases for models to reflect over. GPT-3.5 and GPT-4 were used for initial experiments: future work will involve expanding results on open source models.

## 4.3   Results

We summarize model performances under each setting in table 4.2. Combining episodic retrieval and reflexion maximizes performance gains by well over **doubling** the performance of zero-shot GPT-4. We condense key findings below:

**Episodic Retrieval works across model sizes, unlike Reflexion.**    We find that the ability to self-reflect effectively is an emergent property of stronger models, consistent with (Shinn et al., 2023; Chen et al., 2023). However, both semantic and episodic retrieval are still effective, with episodic retrieval even causing **GPT-3.5 to approach GPT-4's zero-shot performance**. This is likely because self-reflection relies on the internal model's strength to reason over sparse, binary reward signals. Retrieval, on the other hand, allows models to reference existing reasoning and code snippets, requiring less intrinsic model capabilities. Our findings thus corroborate Li et al. (2023a), where LMs can understand much more complex competitive programming solutions than they can produce.

**Episodic Retrieval and Reflexion have strong synergy.**    Episodic Retrieval reaches new maximums when combined with Reflexion, but not with Semantic Retrieval. We find that for GPT-4, **70.2%** of newly solved problems (relative to zero-shot) with semantic retrieval are also newly solved by episodic retrieval. It provides one possible explanation as to why combining the two may decrease performance: the additional knowledge provided by our implementation of semantic retrieval trades off against its long contexts, which current LLMs are known to be struggle with (Liu et al., 2024). In contrast, only **45.9%** of newly solved problems with Reflexion overlap with episodic retrieval, pointing to better synergy.

**Platinum problems remain unsolved.**    Although solve rates on gold problems grow significantly, platinum problems remain unsolved, posing an open challenge for future inference techniques and foundation models. For more details, refer to Appendix C.2

**Performance gains are not due to memorization.**    A competing hypothesis for the success of retrieval is that adding retrieved solutions increases memorization effects for the problem currently being evaluated, rather than the model critically engaging with the content of the retrieved content itself. To test for this, we remove critical sections of retrieved solutions and find significant drops in performance. In addition, qualitative analysis indicates no significant overlap in generated and official published solutions. Full experiment details can be found in Appendix C.1.

## 4.4   Qualitative Analysis

We focus our qualitative analysis on the improvements of retrieval, our individually strongest inference-time technique. Here, we isolate 3 examples of problems newly solvable with RAG in figure 4.

**Example 1, Sample Bronze Problem: 187_bronze_find_the_cow!**

```
# Iterate though grass string from...
for i in range(len(grass)-1, -1, -1):
    if grass[i:i+2] == "))":
        count_front_legs += 1
    elif grass[i:i+2] == "((":
        count_bessite += count_front_legs
        ...
```

```
for i in range(1, len(grass)):
    if grass[i] == '('
    and grass[i-1] == '(':
        back_legs_count += 1
    elif grass[i] == ')'
    and grass[i-1] == ')':
        possible_locations +=
        back_legs_count
        ...
```

```
Perhaps the simplest definition is
that there must be
the same total number of ('s and )'s,
and for any prefix of the string,
there must be at least as many ('s as
)'s.  For example, the following
strings are all balanced:
        ()
       (())
     ()(()())
        ...
```

**Incorrect** zero-shot solution    **Correct** Solution with Problem Memory    Memory: 188_bronze_typo

---

**Example 2: Sample Silver Problem: 210_silver_milk_routing**

```
def milk_routing(n, m, x, pipes):
    graph = {i: [] for i in range(1, n+1)}
    for i, j, l, c in pipes:
        graph[i].append((j, l, c))
        graph[j].append((i, l, c))
    visited = {i: 0 for i in range(1, n+1)}
    while q:
        ...
```

```
def dijkstra(graph, source, n, capacity):
    time_taken = [float('inf')] * (n+1)
    time_taken[source] = 0
    min_heap = [0, source]
    while min_heap:
        ...
```

```
def dijkstra(n, edges, source, destination...
    dist = [float('inf')] * (n+1)
    visited = [(0, source)]
    heapq.heapify(visited)
    dist[source] = 0
    while visited:
        ...
```

**Incorrect** zero-shot solution    **Correct** Solution with Problem Memory    Memory: 969_gold_milk_pumping

---

**Example 3: Sample Gold Problem: 400_gold_roadblock**

```
def main():
    graph, N, M = read_input()
    shortest_distance, shortest_path = dijkstra(..
    max_increase = 0
    for u, v in shortest_path:
        original_length = graph[q][v]
        graph[u][v] *= 2
        ...
```

```
def main(fields, pathways):
    graph = {i: {} for i in range(1,fields+1)}
    for a, b, length in pathways:
        graph[a][b] = length
        graph[b][a] = length
    shortest_path, min_dist = dijkstra(graph...)
    max_increase = 0
    for i in range(1, len(shortest_path))
        ...
```

```
        Let $G$ be a tree.
For every query of the form $(u, v)$
  we want to find the lowest common
  ancestor of the nodes $u$ and $v$,
i.e. we want to find a node $w$ that
lies on the path from $u$ to the root
 node, that lies on the path from $v$
   to the root node, and if there are
multiple nodes we pick the one that is
  farthest away from the root node.
              ...
```

**Incorrect** zero-shot solution    **Correct** Solution with Semantic Memory    Memory: LCA Algorithms

Figure 4: Three examples of problems previously unsolved, but solved with retrieval.

**Example 1: Models can borrow reasoning about similar problem environments**   Here, both the central problem and the most relevant retrieved problem requires models to parse and operate over parenthesis-only strings. The retrieved solution + code thus provides models with sample reasoning over this tricky and mistake-prone problem environment, allowing models to generate more accurate code.

**Example 2: Models can adopt existing code structure and algorithms**   In this example, both problems require the use of Dijkstra's shortest path algorithm. However, the initial implementation contains many small, nuanced bugs due to its attempts to push the entire solution into a single function. This not only makes the code less modular, but it also leaves greater room for error. With episodic retrieval, we see the model grounding itself in the retrieved content, borrowing significant code structure and algorithmic reasoning of the retrieved problem.

**Example 3: Models can utilize algorithmic concepts and reasoning from texts**   The given problem here requires finding a shortest path within a grid containing roadblocks. We see that the model fetches a textbook chapter on lowest common ancestor algorithms, which covers tangentially applicable graph traversal techniques to the problem. Interestingly enough, it does not fetch the chapter on Dijkstra's algorithm for shortest paths, the algorithm utilized in the official problem analysis. Visual inspection of the Dijkstra's chapter indicated that the chapter was short and light on details, thus receiving a low retrieval score. This highlights the flexibility of the retrieval query to adapt to low quality documents, finding suitable replacements.

# 5   Human-in-the-loop Guidance

In benchmark evaluations, we found a wide diversity in the distribution of model errors: from problem misunderstandings, to subtle off-by-one implementation issues. To further examine how far a model is from solving a given problem, we perform a human study via an interactive "tutoring setup."

**Setup**    A human provided with problem solutions engages in a multi-turn exchange with a model, at each step providing feedback on mistakes under a specified ruleset. Notably, the human is not allowed to provide specific fixes rather only general instructions (e.g., "you are using the wrong algorithm" or "this code may result in index out of bounds"). The full interaction ruleset is detailed in Appendix D.1. The model is then prompted to fix its mistakes: we allow a maximum of 1 code execution to simulate a pass@1 setup, a maximum of 5 code generations to limit conversation length, and a total of 3 attempts per problem.

**Results**    Surprisingly, we find that on a small set of 15 problems[1] on which GPT-3.5 and GPT-4 achieve zero pass rate using all of the above inference-time methods, the human-in-the-loop setup raises GPT-4 performance from 0% to 86.7% (13 problems solved) while not improving GPT-3.5 performance from 0%. Qualitatively, GPT-3.5 consistently hallucinates fixes irrelevant to feedback, while GPT-4 pinpoints accurate algorithmic fixes in response to minimalistic feedback. A sample trajectory is provided in Appendix D.2.

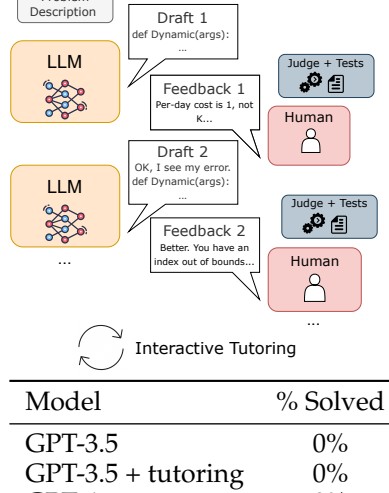

**Problem-level Analysis**    We found that GPT-4 was more responsive to blanket feedback that its algorithm or understanding of an environment concept was incorrect, and more capable of landing on the correct strategy in its second attempt, whereas GPT-3.5's retries are typically similarly unhelpful. For example, GPT-4 lands on the correct overall solution strategy after being instructed to not use a heap in "Hungry Cow," and similarly after being instructed not to use DP in "The Lost Cow." In "Photoshoot," it suffices to ask GPT-4 "Is there any way we can use the inherent ordering of the cows and directly calculate the number of steps necessary?" to correct it from a simulation to a precomputed parity-counting approach; explaining this to GPT-3.5 does not yield any progress. Similarly, GPT-4 is also able to follow instructions on spe-

| Model | % Solved |
|---|---|
| GPT-3.5 | 0% |
| GPT-3.5 + tutoring | 0% |
| GPT-4 | 0% |
| **GPT-4 + tutoring** | **86.7%** |

Figure 5: Human-in-the-loop interactive "tutoring" setup: GPT-4 successfully incorporates feedback while GPT-3.5 does not.

cific fixes, unlike GPT-3.5. For example, on "The Lost Cow," GPT-4 tends to calculate the next position using the current position instead of the *initial* position – if given this fact, it produces the correct implementation. [2]

**Discussion**    Our human-in-the-loop results highlight that the full capabilities of models may not be captured by solve rate; among two models failing on a given problem, one may be one correction away from a fully correct solution, whereas the other may fundamentally misunderstand the problem scenario. This motivates better evaluation metrics beyond execution success (pass@k). Another perspective on our results is that GPT-4 has further reasoning capabilities that may be "unlocked" by human-level corrective feedback, highlighting the need for better methods to generate such feedback.

## 6   Conclusion

In this paper, we introduce the USACO benchmark for rigorously evaluating code language models on tasks involving grounded ad hoc reasoning and novel algorithmic thinking. We observe that foundation models previously shown to excel at basic coding tasks like HumanEval (Chen et al., 2021) perform poorly zero-shot in these more challenging scenarios,

---

[1]13 bronze, 1 silver, 1 gold. We focused on bronze problems since models performed poorly on harder difficulties.

[2]See https://benshi34.github.io/blog/2024/human-in-the-loop/ for more trajectories and analysis for different difficulty levels.

but that providing models with task-specific knowledge stores can well over double zero-shot performance. We hope that our evaluation of current models' limitations and findings on the effectiveness of semantic and episodic knowledge help lay groundwork for its integration into future models and language agents alike.

## 7  Reproducibility

We release all data and code at https://princeton-nlp.github.io/USACOBench/, as well as human in the loop trajectories at https://benshi34.github.io/blog/2024/human-in-the-loop/. We advise others to use isolated execution environments when reproducing experiments as the generated code is not validated before execution.

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

# A    Evaluating with a train-test split

As described in Section 4.2, our episodic retrieval evaluation setting involves holding out one problem at a time (i.e., retrieves from the solutions of all other test problems), with the intention of maximally capturing the effect of retrieval on the relatively small dataset available. We expect that this does not lead to any significant dataset leakage across evaluation, as problems are highly independent and ad hoc, with little solution logic shared by even problems of the same algorithm type.

| Technique | GPT-3.5 | GPT-4 |
|---|---|---|
| Zero-Shot (7B) | 0.795 | 8.87 |
| Reflexion (7B) | 0.89 | 11.92 |
| Episodic Retrieval | 2.65 | 9.93 |
| Reflexion + Episodc Retrieval | **4.64** | **14.57** |

Table 4: A standard train-test split shows similar results across inference-time techniques, compared to our leave-one-out episodic retrieval setting.

Nonetheless, we re-ran the majority of the inference-time techniques against a setting with a more standard train-test split. We standardize problem difficulty by generating the same difficulty distribution as the entire USACO set. As shown in Figure A, as expected, the standard split (train n=200, test n=107) yields similar conclusions with slightly reduced retrieval effectiveness. This is because the amount of problems retrieved over is reduced, thus on average decreasing problem similarity between retrieved problems and the problem currently being solved. Additionally, we re-tune p (number of retrieved passages) on this train set only, and recover the same optimal values as the leave-one-out setting.

# B    USACO Details

## B.1    USACO Zero-Shot Prompt

```
Please reply with a Python 3 solution to the below problem. Make sure to wrap your code in
'```python' and '```' Markdown delimiters, and include exactly one block of code with the
entire solution (in the final code step).
Reason through the problem and think step by step. Specifically:
1. Restate the problem in plain English
2. Conceptualize a solution first in plain English
3. Write a pseudocode solution
4. Output the final Python solution with your solution steps in comments.
No outside libraries are allowed.

[BEGIN PROBLEM]
{INSERT PROBLEM HERE)
[END PROBLEM]
```

Figure 6: USACO zero-shot prompt. We find that asking the model to describe its reasoning + pseudocode first lead to more parse-able and human-interpretable generations, making qualitative analysis greatly simplified. It is also beneficial for utilization in retrieval queries, as it contains more algorithmic keywords. However, it does not boost the performance significantly: the choice to prompt it to do the extra steps for zero-shot performance evaluation is more for interpretability benefits.

## B.2   USACO Zero-Shot Error Breakdown

| Model | Wrong Ans. | TLE | MLE | Runtime | Syntax + Other |
|---|---|---|---|---|---|
| CodeLlama (7B) | 61.10 | 5.05 | 0 | 9.89 | 23.85 |
| DeepSeek Coder (7B) | 80.62 | 10.04 | 0 | 6.82 | 1.86 |
| Claude-3-Sonnet | 55.70 | 24.42 | 0 | 14.66 | 2.61 |
| GPT-3.5 | 77.42 | 7.02 | 0 | 13.37 | 1.82 |
| GPT-4 | 44.15 | 38.20 | 0 | 10.41 | 1.18 |

Table 5: Error distributions zero-shot, in %. TLE indicates time limit exceeded, and MLE indicates memory limit exceeded. "Other" generally represents errors stemming from models outputting incorrectly formatted code.

## B.3   USACO Zero-Shot Performance By Difficulty

| Model | Bronze | Silver | Gold | Platinum |
|---|---|---|---|---|
| CodeLlama (7B) | 0.41 | 0 | 0 | 0 |
| DeepSeek Coder (7B) | 2.30 | 0 | 0 | 0 |
| GPT-3.5 | 1.46 | 0 | 0 | 0 |
| Claude-3-Sonnet | 5.69 | 1.00 | 0 | 0 |
| GPT-4 | **19.11** | **3.10** | **0.16** | 0 |

Table 6: Zero-shot performance of models by difficulty

### B.4 USACO Qualitative Analysis

| Problem ID | Problem overview | GPT-4 error |
|---|---|---|
| 1277_bronze_moo_operations | Perform string operations to change a string to "MOO" | **(problem understanding)** Arbitrarily manipulates letters, doesn't progress toward target |
| 1300_bronze_stamp_grid | Check if copies of a 2D array can be "stamped" to produce a larger 2D array | **(format issue)** Doesn't account for empty line in input |
| 1275_bronze_leaders | Count number of "leaders" in a list, following a definition | **(problem understanding)** Arbitrarily counts, doesn't adhere to definition |
| 1301_bronze_watching_moooo | Choose cost-minimizing days to buy subscriptions | **(wrong approach)** Implements faulty dynamic programming instead of sorting and iterating |
| 735_bronze_the_lost_cow | Move in a pattern along a 1D number line until colliding with a target | **(problem understanding)** Forgets to jump back to the beginning after each pattern iteration |

Figure 7: An overview of some common, general errors on bronze problems, as easier problems generally have only 1-2 points of failure and is simpler to analyze.

## B.5 Sample Generation

```python
# Restating the problem in plain English:
# - Farmer John is at position x and Bessie is at y on the same line.
# - Farmer John doesn't know where Bessie is and must find her by walking back and forth.
# - He will walk in a zig-zag pattern: first to x + 1, then to x - 2, then to x + 4, and so on.
# - Each leg of the journey is twice as long as the previous one and in the opposite direction.
# - The task is to calculate the total distance Farmer John will walk until he finds Bessie.

# Conceptualizing a solution first in plain English:
# - Farmer John alternates directions, first adding 1, then subtracting 2, adding 4, etc.
# - We track the total distance walked and the current position until Farmer John passes position
y.
# - We need to take into account both the distance to reverse directions and the distance walked
towards y.
# - We'll increment a step size that starts at 1 and doubles each step, alternating addition and
subtraction.

# Pseudocode solution:
# 1. Initialize total_distance and current_position to 0.
# 2. Initialize direction to 1 (as Farmer John first moves to position x+1).
# 3. Initialize step to 1.
# 4. While Farmer John has not passed position y:
#     a. Increment total_distance with the absolute of step.
#     b. Update current_position by adding step.
#     c. Change direction (if it was 1, make it -1, otherwise make it 1).
#     d. Double the step value and multiply it by direction for the next iteration.
# 5. Since Farmer John would have passed y, add the distance from position y to where he passed.
# 6. Print the total_distance.

# Output the final Python solution with solution steps in comments.
x, y = map(int, input().split())

total_distance = 0  # Total distance traveled
current_position = x  # Current position of Farmer John
direction = 1  # Direction Farmer John is moving, 1 for right, -1 for left
step = 1  # Step size, will double each time

# Continue zig-zagging until Farmer John passes Bessie's position
while (direction == 1 and current_position < y) or (direction == -1 and current_position > y):
    # Walk in the current direction by the current step size
    total_distance += step
    # Update the current position
    current_position += step
    # Reverse the direction for the next step
    direction *= -1
    # Double the step for the next iteration
    step *= 2

# Farmer John walked past Bessie's position, so subtract the overshoot distance
overshoot = abs(current_position - y)
total_distance -= overshoot

# Output the total distance traveled
print(total_distance)
```

Figure 8: Sample zero-shot generation by GPT-4

# C   Inference Time Methods

## C.1   Addressing Memorization

We address memorization by performing ablations on the corpus in table C.1: if retrieved problem solutions were spurring regurgitation of memorized solutions to the current problem, removing core parts of the retrieved solutions should not dwindle this effect. However, we find that it does: utilizing solely the problem description only retains 3.18% of the performance, suggesting that models are truly utilizing the reasoning of similar problems provided in context to inform their generations.

| Retrieval Content | % of maximum performance |
|---|---|
| **PD** | **3.18** |
| PD + Code + English Solution | 100 |

Table 7: Ablations on episodic corpus content. PD represents problem description.

## C.2   Per Difficulty Pass Rates

| Model | Bronze | Silver | Gold | Platinum |
|---|---|---|---|---|
| Reflexion | 24.39 | 5 | 4.76 | 0 |
| Semantic Retrieval | 19.72 | 6.0 | 1.98 | 0 |
| Episodic Retrieval | 26.22 | 9.5 | 3.6 | 0 |
| Episodic Retrieval + Reflexion | 35.77 | 14.0 | 6.35 | 0 |

Table 8: GPT-4 Pass@1 rates for inference methods per difficulty: We report incrementally superior methods.

## C.3   Retrieval Query Ablations

| Query Content | Performance |
|---|---|
| PD | 13.36 |
| PD + Attempted Code Solution | 14.33 |
| PD + Attempted English + Code Solution | 14.98 |

Table 9: Ablations on retrieval query, PD represents problem description. Performance measured in pass@1. We find that generally most retrieval queries are somewhat effective, however, including code attempts as well as english solution performs the best, as it allows the maximum matching of relevant keywords between compared documents.

## C.4 Environment duplicates

```
1276_bronze_air_cownditioning_ii   :   ['1156_bronze_air_cownditioning',      395_bronze_auto-complete   :   ['397_silver_auto-complete', '897_gold_cow_poetry']
'346_gold_empty_stalls']                                                       379_silver_bessie_slows_down   :   ['377_bronze_bessie_slows_down', '989_bronze_race']
1251_bronze_cow_college   :   ['1155_bronze_lonely_photo',                     377_bronze_bessie_slows_down   :   ['379_silver_bessie_slows_down', '989_bronze_race']
'1035_bronze_social_distancing_i']                                             361_silver_milk_scheduling   :   ['246_silver_milk_scheduling', '832_bronze_milking_order']
1204_bronze_photoshoot_2   :   ['1183_silver_cow_frisbee',                     360_bronze_wormholes   :   ['137_gold_tied_down', '243_bronze_perimeter']
'1209_gold_redistributing_gifts']                                             342_bronze_farmer_john_has_no_large_brown_cow   :
1017_gold_timeline   :   ['1069_platinum_spaceship', '866_platinum_the_cow_gathering']   ['343_silver_farmer_john_has_no_large_brown_cow', '433_silver_fair_photography']
1011_bronze_triangles   :   ['1015_silver_triangles', '641_bronze_field_reduction']   259_bronze_cow_race   :   ['205_bronze_meet_and_greet', '377_bronze_bessie_slows_down']
965_bronze_livestock_lineup   :   ['832_bronze_milking_order',                 246_silver_milk_scheduling   :   ['361_silver_milk_scheduling', '832_bronze_milking_order']
'361_silver_milk_scheduling']                                                 241_bronze_message_relay   :   ['281_bronze_haywire', '361_silver_milk_scheduling']
944_silver_fence_planning   :   ['642_silver_field_reduction',                 242_bronze_cow_crossings   :   ['433_silver_fair_photography', '241_bronze_message_relay']
'1040_silver_the_moo_particle']                                              228_silver_party_invitations   :   ['433_silver_fair_photography',
939_bronze_bucket_brigade   :   ['243_bronze_perimeter', '380_silver_cross_country_skiing']   '96_bronze_escaping_the_farm']
941_bronze_cow_evolution   :   ['893_bronze_guess_the_animal',                 225_bronze_liars_and_truth_tellers   :   ['261_bronze_breed_assignment',
'1281_gold_find_and_replace']                                                 '243_bronze_perimeter']
916_bronze_the_great_revegetation   :   ['894_silver_grass_planting',          210_silver_milk_routing   :   ['969_gold_milk_pumping', '415_silver_watering_the_fields']
'1181_bronze_drought']                                                       205_bronze_meet_and_greet   :   ['259_bronze_cow_race', '967_silver_meetings']
894_silver_grass_planting   :   ['916_bronze_the_great_revegetation',          193_gold_balanced_cow_breeds   :   ['192_silver_balanced_cow_breeds',
'120_gold_nearby_cows']                                                       '194_gold_concurrently_balanced_strings']
891_bronze_shell_game   :   ['893_bronze_guess_the_animal', '833_bronze_family_tree']   191_silver_distant_pastures   :   ['283_silver_fuel_economy', '861_gold_fine_dining']
892_bronze_sleepy_cow_sorting   :   ['898_gold_sleepy_cow_sorting',            192_silver_balanced_cow_breeds   :   ['193_gold_balanced_cow_breeds',
'918_silver_sleepy_cow_herding']                                              '194_gold_concurrently_balanced_strings']
860_silver_mooyo_mooyo   :   ['243_bronze_perimeter', '1212_platinum_paint_by_rectangles']   187_bronze_find_the_cow!   :   ['188_bronze_typo', '193_gold_balanced_cow_breeds']
835_silver_lemonade_line   :   ['808_bronze_hoofball', '898_gold_sleepy_cow_sorting']   130_bronze_cows_in_a_row   :   ['229_gold_cow_lineup', '260_bronze_breed_proximity']
831_bronze_team_tic_tac_toe   :   ['378_bronze_balanced_teams', '1327_silver_field_day']   118_gold_cow_coupons   :   ['103_bronze_gifts', '99_silver_umbrellas_for_cows']
788_silver_mootube   :   ['789_gold_mootube', '1037_bronze_cowntact_tracing']   107_silver_bale_share   :   ['243_bronze_perimeter', '244_silver_perimeter']
784_bronze_lifeguards   :   ['786_silver_lifeguards', '209_silver_wifi_setup']   103_bronze_gifts   :   ['118_gold_cow_coupons', '1209_gold_redistributing_gifts']
767_gold_haybale_feast   :   ['862_gold_cowpatibility',                        98_silver_roadblock   :   ['398_silver_roadblock', '400_gold_roadblock']
'1188_platinum_minimizing_haybales']                                          96_bronze_escaping_the_farm   :   ['839_gold_talent_show', '1161_gold_paired_up']
762_silver_my_cow_ate_my_homework   :   ['648_platinum_262144', '812_silver_teleportation']   89_silver_cow_lineup   :   ['229_gold_cow_lineup', '260_bronze_breed_proximity']
549_silver_bessie_goes_moo   :   ['546_bronze_bessie_gets_even',
'812_silver_teleportation']
400_gold_roadblock   :   ['398_silver_roadblock', '98_silver_roadblock']
397_silver_auto-complete   :   ['395_bronze_auto-complete', '1205_bronze_blocks']
398_silver_roadblock   :   ['98_silver_roadblock', '400_gold_roadblock']
394_bronze_mirror_field   :   ['223_bronze_mirrors', '244_silver_perimeter']
```

Figure 9: A list of problems that were not originally solved zero-shot, but is solved with episodic retrieval, as well as a list of the relevant problems retrieved by problem id, listed in decreasing order of relevancy.

**Problem environment duplicate retrieval**    We want to quickly point out here that USACO reuses problem environments occasionally. For example, there exists both gold and platinum versions of the problem "pareidolia," both asking users to develop different algorithms regarding to strings. Although the problems utilize the same problem environment, the algorithms and reasoning behind the solutions differ greatly, making it still a nontrivial task to solve the problem even given the solutions to the alternate problem. Additionally, we find that only 33% of newly solved problems contain one or more retrieved content that fall into this category: the full list can be found in figure 9. The rest retrieve problems that are completely separate.

## C.5 Hyperparameter Tuning

| Problems Retrieved | Pass@1 |
|:---:|:---:|
| $p = 1$ | 13.03 |
| $p = 2$ | 14.33 |
| $p = 3$ | 13.11 |
| $p = 4$ | 12.38 |

Table 10: Episodic retrieval hyperparameter tuning: Here we tune over how many problems to retrieve over on the USACO307 dataset. GPT-4-turbo-1106 was used in all experiments here. We see that $p = 2$ is optimal in pass@1. We did not test resampling for greater numbers of $p$ to conserve budget as performance in pass@1 was already dropping.

| Reflexion Iterations | Pass@i |
|:---:|:---:|
| $i = 0$ | 8.7 |
| $i = 1$ | 10.75 |
| $i = 2$ | 12.28 |
| $i = 3$ | 12.38 |
| $i = 4$ | 12.38 |
| $i = 5$ | 12.40 |

Table 11: Reflection iteration tuning: Here we tune over how many times to iterate. All experiments were done with GPT-4. $i = 0$ indicates the original solve rate without any reflection. We see that solve rates remain relatively static after 3 iterations.

## C.6 Reflexion Prompt

```
You were previously solving a coding problems. Here is the problem that you were solving:
    {problem_dict[query['problem_id']]['description']}
    And here are all your past attempts, as well how your code fared on the unit tests for the problem:
    {query['reflection_buffer']}
    Think carefully about where you went wrong in your latest solution, first outputting why you think
you went wrong. Then, given your insights, try to fix the solution, outputting a block of correct
python3 code to be executed and evaluated again. Make sure to wrap your code in '```python' and '````'
Markdown delimiters.
```

Figure 10: Reflexion prompt

## C.7 Retrieval Prompts

```
Please reply with a Python 3 solution to the below problem. Make sure to wrap your code in '```python' and '````'
Markdown
delimiters, and include exactly one block of code with the entire solution
(in the final code step). You will also be given multiple somewhat similar problems, as well as the solution to those
similar problems. Feel free to use those problems to aid your problem solving process.
Reason through the problem and:
1. Restate the problem in plain English
2. Conceptualize a solution first in plain English
3. Write a pseudocode solution
4. Output the final Python solution with your solution steps in comments.
No outside libraries are allowed.

[BEGIN SIMILAR PROBLEMS]
{query['retrieval_text']} (Similar problem problem + solution goes here)
[END SIMILAR PROBLEMS]

Now it's your turn. Here is the problem you are to solve:
[BEGIN PROBLEM]
{problem_dict[query['problem_id']]['description']} (Description of problem goes here)
[END PROBLEM]"""
```

Figure 11: Episodic Retrieval Prompt

```
Please reply with a Python 3 solution to the below problem. Make sure to wrap your code in '```python' and '```'
Markdown
delimiters, and include exactly one block of code with the entire solution
(in the final code step). You will also be given a textbook section relevant to the given problem. Feel free to use
that to aid your problem solving process.
Reason through the problem and:
1. Restate the problem in plain English
2. Conceptualize a solution first in plain English
3. Write a pseudocode solution
4. Output the final Python solution with your solution steps in comments.
No outside libraries are allowed.

[BEGIN RELEVANT TEXTBOOK SECTION]
{query['retrieval_text']} (Textbook chapter goes here)
[END RELEVANT TEXTBOOK SECTION]

Now it's your turn. Here is the problem you are to solve:
[BEGIN PROBLEM]
{problem_dict[query['problem_id']]['description']} (Description of problem goes here)
[END PROBLEM]"""
```

Figure 12: Semantic Retrieval Prompt

```
You were previously solving a coding problems. Here is the problem that you were solving:
    {problem_dict[query['problem_id']]['description']}
    You were also given a couple of similar problems to the problem above along with their solutions to
aid you in solving the problem at hand. Here are the similar problems you were given:
    {query['retrieval_text']}
    And here was your original response.
    {query['original_response']}
    Here was the execution result of the above solution:
    {query['execution_response']}
    Think carefully about where you went wrong. Then, try to fix the solution, outputting a block of
correct python3 code to be executed and evaluated again. Make sure to wrap your code in '```python' and
'```' Markdown delimiters.
```

Figure 13: Reflexion + Retrieval Prompt

# D Human-in-the-loop

## D.1 Human-in-the-loop Prompt

Human Tutoring Instructions

You are to interact with a given model to try to solve a given coding question. A problem solving session ends whenever the model has generated code 5 times. Between code generations you may speak to the model in conversation as many times as you would like. However, the way you interact with the model must be very specific: your goal is to act akin to a tutor and guide the model towards the right answer, without giving away any underlying details about the true solution of the problems.

You MAY NOT provide the models with:
- The correct algorithm to fix their initial solution
- Exact lines of code where they went wrong
- Explanations of the problem or explanations of misunderstandings

You MAY provide the models with:
- General concepts/data structures that may be useful for solving the problem
- Walking through a sample input output of the problem to better verify problem understanding
- Short concise general directions on where the code went wrong

Here are examples of correct trajectories:

Figure 14: The prompt given to human tutors when interacting with the model. We manually inspect trajectories to ensure rules were followed. To limit the amount of conversation, we only allow the model to generate code 5 times: however, the code is only allowed to be executed once to simulate a pass@1 setup.

## D.2 Human-in-the-loop sample trajectory

Please reply with a Python 3 solution to the below problem. Make sure to wrap your code in `'python'` and `''` Markdown delimiters, and include exactly one block of code with the entire solution. Feel free to either return just the one code block with your solution or the one code block with explanatory text before and/or after -- however, you will only be evaluated on the correctness of your code.

[BEGIN PROBLEM] Farmer John has lost his prize cow Bessie, and he needs to find her!

Fortunately, there is only one long path running across the farm, and Farmer John knows that Bessie has to be at some location on this path. If we think of the path as a number line, then Farmer John is currently at position $x$ and Bessie is currently at position $y$ (unknown to Farmer John). If Farmer John only knew where Bessie was located, he could walk directly to her, traveling a distance of $|x - y|$. Unfortunately, it is dark outside and Farmer John can't see anything. The only way he can find Bessie is to walk back and forth until he eventually reaches her position.

```python
def lost_cow(x: int, y: int) -> int:
    direction = 1
    start_position = x
    total_distance_traveled = 0

    while True:
        destination = start_position + direction
        total_distance_traveled += abs(start_position - destination)

        if min(start_position, destination) <= y <= max(start_position, des
            break

        direction *= -2
        start_position = destination

    return total_distance_traveled
```

This code is not correct. There are several bugs in your code, but I think it's important to verify your understanding of the problem. Let's converse and make sure you understand the problem correctly. Let's go through the sample case together, where the farmer starts at 3, and the cow is at 6. After 1 round, where should the farmer be?

If Farmer John starts at position 3, in his first move, he will shift to position 3+1=4. Then he changes his direction and moves twice the distance he moved in his previous step, so he moves 2 steps towards the left from his current position to end up in position 4-2=2. Note that while his current position is 2, he has traveled a total of 3 units - 1 unit to the right and 2 unit to the left.

Okay. So just to verify you understand, where should he be in round 3?

In round 3, Farmer John changes his direction again and now moves twice the distance he moved in the previous step i.e., 4 units to the right from his current position of 1. So, he will end up at position 1 + 4 = 5. This round covers a total distance of 4 units. So after 3 rounds, he should be at position 5.

This is not correct. He should actually be at position 3 + 4 = 7. What are you missing in your understanding of the problem?

My mistake. I misunderstood the way direction changes and distance increments were accounted for.

In each round, when Farmer John reverses direction, he doesn't move in relation to his current position. Instead, he moves relative to his starting position, 'x'. In each round, the distance he moves doubles in magnitude and alternates in direction.

Yes this is correct! Each movement is relative to his starting position. Knowing this, can you implement code that exhibits the intuition we described above?

```python
def lost_cow(x: int, y: int) -> int:
    current = x
    direction = 1
    total_distance_traveled = 0

    while True:
        # Move to the next position relative to starting position
        next_position = x + direction

        # Accumulate total distance traveled
        total_distance_traveled += abs(next_position - current)

        # If Bessie is found between old and new positions, reduce
        # excess travel distance and break loop
        if min(current, next_position) <= y <= max(current, next_position):
            total_distance_traveled -= abs(next_position - y)
            break

        # Prepare for next iteration
```

Figure 15: Sample human tutoring trajectory on the problem 786_bronze_the_lost_cow

