# OpenReview forum: "Can Language Models Solve Olympiad Programming?"
_colmweb.org/COLM/2024/Conference — COLM_

### Official Review · Reviewer_mWB9 · 2024-05-12

**Rating:** 8
**Confidence:** 4
**Ethics Flag:** 1

**Summary:**

This paper introduces USACO, a coding benchmark with 307 challenging problems from past USA Computing Olympiad competitions. These problems require a wide range of algorithmic, mathematical, and commonsense knowledge, as well as grounded reasoning. Even the best LLM (i.e. GPT-4) only achieves a zero-shot pass@1 solve rate of 8.7% on USACO, indicating the need for better models and inference methods. To address this, high-quality unit tests, reference code solutions, and official analysis for each problem, along with instructional texts in the form of competition programming textbooks are collected.  Baseline methods based on retrieval and self-reflection, and combined are proposed to improve performance. However, all methods still fall short of solving the benchmark above the bronze level. To gain further insights, a human-in-the-loop study where participants interact with the language models in a tutoring setup is further conducted. The study highlights the potential of incorporating high-quality feedback and the need to develop methods for generating human-level corrective feedback.

**Questions To Authors:**

I am wondering if you have any measures to alleviate the data contamination issue, as the problems and the ground-truth solution are collected from the public-facing pages [1][2]. Researchers may also crawl the problems and the corresponding solutions from these websites to pre-train or instruction tune the LLMs. Prior efforts adopted different measures to deal with this case. For example, DS1000 rewrote problems and reference solutions. Another one is codecontests, all training data in CodeContests was publicly released on or before 2021/07/14. The test set contains problems published after 2021/09/21.

References:
[1] Stop Uploading Test Data in Plain Text: Practical Strategies for Mitigating Data Contamination by Evaluation Benchmarks. EMNLP 2023
[2] NLP Evaluation in trouble: On the Need to Measure LLM Data Contamination for each Benchmark. EMNLP 2023
[3] Competition-level code generation with alphacode. Science 2022
[4] DS-1000: A Natural and Reliable Benchmark for Data Science Code Generation. ICML 2023

**Reasons To Accept:**

I think this is a nice resource paper. The proposed benchmark USACO has the potential to become a valuable tool for advancing research in program synthesis with LLMs.

- Well-written and organized: The paper is clear and easy to follow, with a logical structure that guides the reader through the research.
- Motivation: This paper effectively highlights the limitations of existing benchmarks and the need for a more challenging and realistic evaluation of LLM capabilities. The proposed USACO addresses this critical gap in existing function-level code generation benchmarks, which are saturated and may not accurately reflect the real program synthesis abilities of LLMs.
- Comprehensive experiments: This paper presents extensive experiments on the USACO dataset, exploring various inference-time LM methods and conducting a human-in-the-loop study to gain deeper insights into the capabilities and limitations of LLMs for Olympiad programming. The human-in-the-loop study provides valuable insights into the strengths and weaknesses of LLMs for program synthesis, complementing the quantitative results obtained through automatic experiments.

**Reasons To Reject:**

I do not have any major concerns about this paper, but I do have a few questions about the dataset construction and data contamination  (see below).

---

> ### Author Rebuttal · Authors · 2024-05-31
>
> Thank you for your encouraging and thoughtful comments!
>
> Regarding your question about potential data contamination: We are optimistic that the **ongoing** nature of the USA Computing Olympiad contests will help make the USACO benchmark leakage-resistant by allowing us to continuously update the benchmark with new contest problems. We (1) hope to release updated versions of the benchmark that include newer contest problems (2) have designed our benchmark to be easy to use in ways that both reveal and resist leakage, including standardized problem and solution formats that simplify supporting new and custom problems, as well as easy-to-use and efficient batch evaluation infrastructure that can reveal temporal effects by comparing performance on subsets of the benchmark data.
>
> As an example of the second point, in the paragraph of Section 3.1 in the paper entitled "Temporal effects are strong," we discuss an experiment that additionally evaluates on a subset of 36 problems released after training cutoffs and find that solve rates reduce, pointing to potential contamination in current models even while they perform poorly on the vast majority of the benchmark.
>
> On problem rewriting: the high difficulty of USACO problems makes them relatively infeasible to rewrite compared to DS-1000 [1]. Simultaneously, though, that same difficulty leads models to struggle on even potentially-leaked older problems. Thus, **beyond mitigating leakage by updating with new problems, we expect the current benchmark to be resistant to effects from leakage and remain a useful, challenging code evaluation.**
>
> [1] DS-1000: A Natural and Reliable Benchmark for Data Science Code Generation. ICML 2023

---

> > ### Comment · Reviewer_mWB9 · 2024-06-04
> >
> > Thanks for the response. I think the authors addressed some of my concerns. I will keep my score at 8.

---

### Official Review · Reviewer_hDmn · 2024-05-12

**Rating:** 7
**Confidence:** 4
**Ethics Flag:** 1

**Summary:**

The authors present a new benchmark with 300 competitive programming exercises, which unlike prior benchmarks combines high-quality unit tests (for automatic evaluation), expert-written analyses of the problems (for high-level analysis), a wide range of highly-applied problem statements (to stress the ability of LMs to write code that adapts to novel problem settings). The nature of these competitive programming exercises lends them well to evaluation: by design, each problem is assigned a (human) difficulty level from bronze to platinum as well as realistic limits on run time and memory constraints. The authors evaluate several LMs within different prompt pipelines, and find that they struggle with almost all questions, and generally answer very few of the highest-difficulty questions, despite producing runnable code. The authors conduct several insightful analyses, including a simple human-in-the-loop in which GPT-4 can consistently learn from human interactions to correct its mistakes, while GPT-3.5 can never productively leverage similar feedback.

**Questions To Authors:**

The reporting of pass@1 and pass@4 are inconsistent and justified in an unclear way. The authors say they switch from pass@1 to pass@4 in Figure 3 "fairly compare to the setup only utilizing reflection memory, whose pass-rates stabilize past 3 inclusions of previous attempts". Is such Reflexion getting exposed to any hidden test data? If not, then the system can realistically do as many internal trials as it wants while only "submitting" once, making this arguably just pass@1.

How was the set of 15 problems for the human-in-the-loop study chosen? They appear non-uniform, i.e. 13 bronze, 1 silver, 1 gold. In any case, the authors use their human-in-the-loop study to suggest that we need "better evaluation metrics beyond the overly strict execution success", but as far as I can tell it's plausible that we just need to evaluate pass@K for larger K or possibly Reflexion-like methods, which appear to expose the difference between LMs better than pass@1, alleviating most of this concern. If this is not true, I don't think the current discussions make that clear.

The authors report an intriguing and useful figure: they evaluate LMs on 36 problems released after training cutoff dates and "find that solve rate drops to 0 for all models", noting that "USACO questions are well known to increase in difficulty every year, making this likely an effect of both difficulty increases as well as inclusion in pre-training data". Dropping to zero is concerning. What about pass@4 for the best LM systems tested, for the new "bronze" (easy) questions. Is it still at zero? **If** so, and unless there's a very dramatic increase in difficulty in the competitions, with redefinition of the notion of "bronze" to make them much harder in the new 36 problems, this suggests a very serious amount of leakage in practice to LM training data. That would not be the authors' fault in any way, but it does pose problems for the utility of this data as an ideal benchmark.

**Reasons To Accept:**

The benchmark is well-motivated, well-described, and well-constructed, and I can see it becoming a standard evaluation testbed after other existing coding (and other) benchmarks in this space have been saturated and lost much of their original power to distinguish the best LMs.

The authors do a great job laying out how the unique properties of their data (some by nature of their source; others by construction) allow for a deeper level of investigation into LM abilities, e.g. the difficulty level from bronze to platinum, realistic limits on run time and memory constraints, covering a wide range of highly-applied problem statements.

The findings of LM failures are well-explored and are likely to motivate real and near-term advancements in LM systems that outperform the current generation of systems.

**Reasons To Reject:**

The authors' own evaluation methodology is ambiguous at best. For example, some of their evaluations consider "retrieval" (from *every other* problem-solution pair) to solve problems. It's unclear that this is a reasonable setting, unless the level of independence between problems is very high. Could the authors instead have divided the data by competition year into dev/test for example, and held the test data out to a higher standard?

Along similar lines but perhaps more confusing are the following. Quite a few explorations like Episodic Knowledge Store (e.g., perhaps most egregiously parameter p's tuning in Table 8) are evaluated directly on the 300-example test set in a way that will quickly reduce the shelf-life of this data, and is reminiscent of essentially tuning on the test set. Are some of these meant to represent some sort of oracle settings? I understand the need to explore some hyperparameters, and that's why I advise the authors to dedicate some data explicitly for tuning/development. Of course, this all challenges the usage of "zero-shot" prompts, since some data is now more explicitly available for tuning.

---

> ### Author Rebuttal · Authors · 2024-05-31
>
> We highly appreciate the thoroughness of the review given! We address your concerns below:
>
> ## 1. Evaluation setting
> The retrieval setup is based on:
>
> (1) Qualitatively, problems are indeed **very independent** with little solution logic shared by even problems of the same algorithm type.
>
> (2) Our goal is to **maximally** showcase the effect of retrieval on a relatively *small* dataset using a k-fold-type evaluation that holds out one problem at a time.
>
> For verification, we have added a traditional train (n=200) and test set (n=107), achieving similar conclusions with slightly reduced retrieval effectiveness:
> | Tech. | GPT-3.5 | GPT-4 |
> | -------- | ------- | ------- |
> | Zero-Shot |  0.795 | 8.87 |
> | Reflexion | 0.89 | 11.92 |
> | Episodic | 2.65 | 9.93  |
> | Reflexion + Episodic | **4.64** | **14.57** |
>
> (Certain settings omitted for concision)
>
> We also re-tuned p (num retrieved passages) on this train set only, and got the same values as the original.
>
> ## 2. Reporting Pass@1
> We ran additional experiments replicating Figure 3 **entirely in pass@1**, following the suggested setup of pass@1 Reflexion. The choice of pass@k does not affect our key findings.
> | Tech. | GPT-3.5 | GPT-4 |
> | -------- | ------- | ------- |
> | Zero-Shot |  0.59 | 8.70 |
> | Reflexion | 0.97 | 12.38 |
> | Episodic | 5.49 | 14.33  |
> | Reflexion + Episodic | **8.79** | **20.20** |
>
> ## 3. Human-in-the-loop setup
> The non-uniform distribution across problem difficulties reflects our **focus on bronze problems** which have relatively higher solve rate yet overwhelmingly remain unsolved using current techniques.
>
> Further experiments show pass@k for k=10 improves performance across models, making differences less interpretable. We will update the paper to make this more clear!
>
> ## 4. Time-sensitivity
> We found that even with pass@k for k=4 on bronze problems, performance **still** drops to zero for the set of 36 new problems we tested.
>
> We are indeed concerned about any data contamination. However, it is also useful to keep in mind that:
>
> (1) USACO problem difficulty increases **substantially** year-over-year. Through community difficulty labels (https://codetiger.me/project/usaco/) we see some recent bronze problems compare with the difficulty of old gold problems!
>
> (2) Our benchmark is sufficiently challenging to remain almost entirely unsolved despite potential contamination.
>
> We also hope supporting the continuous addition of new USACO problems can further mitigate contamination.

---

> > ### Comment · Reviewer_hDmn · 2024-06-03
> >
> > Thank you for the discussion. The content does not fundamentally alter my assessment of the work, so I'll keep my 7/10 score.

---

### Official Review · Reviewer_hsvi · 2024-05-20

**Rating:** 8
**Confidence:** 4
**Ethics Flag:** 1

**Summary:**

The paper contributes an important benchmark of challenging problems from the USA Computing Olympiad. These questions are shown to be quite challenging for the current state of the art LLMs including some that for meant specifically for code generation. In addition to the questions, the USACO corpora that is contributed also contains other very useful resources that can help improve the LLM capabilities like exhaustive unit tests and expert written problem analysis.

In addition to the corpora, the also explore improving the zero-shot performance of the LLMs through various different techniques, like (1) Self-reflexion, (2) Retrieval augmentation, both semantic and episodic; and their combinations. They found that looking at it's own output and execution feedback helps a lot. They also see that combining them with episodic augmentation helps the most.

They further showed how the model can improve significantly by a human in the loop framework. This very interesting as it suggests that with the right set of reasoning guidance the model can actually get the answers correctly.

**Questions To Authors:**

1) Among the 15 questions that were chosen for the human trajectory, and used to solve with GPT-4 and GPT-3.5 what was distributions of the 13 questions GPT-4 got right( #s of gold, silver and bronze)?

2) Have you thought about how are you planning on keeping the benchmark updated? As I understand as there are more training runs, it's possible that these benchmarks get consumed in the LLM training. What measures do the authors plan to take to mitigate that?

**Reasons To Accept:**

This paper makes a significant contribution by contributing an important benchmark of challenging programming questions. These evaluations help present gaps in the code understanding the reasoning capabilities.

The authors also do extensive study on how to do well at these tasks and by sharing these approaches it helps get insights into understanding what could potentially be lacking in LLMs and how these capabilities can be enhanced.

The paper is also clearly written and well presented. Including the appendix section which helps the reader understand how the models were prompted.

**Reasons To Reject:**

My biggest concern is how to handle the time machine problem. As the models get trained with more recent data the benchmarks has the potential to become not so useful. It would be great to get a sense of how the authors are planning on keeping the benchmark updated.

---

> ### Author Rebuttal · Authors · 2024-05-31
>
> Thank you for your encouraging comments and suggestions! We address and give relevant clarifications for each of your questions/concerns below:
>
> ## 1. Clarification on human-in-the-loop trajectories
>
> We appreciate the interest in the details of the human-in-the-loop experiment!
>
> Since models showed poor pass@1 performance on difficulties harder than bronze, we focused on bronze problems for the human-in-the-loop experiments, choosing a distribution of **13 bronze, 1 silver, and 1 gold**.
>
> (The strongest model, GPT-4, achieved 19.1% on bronze, 3.1% on silver, and only 0.16% on gold)
>
> ## 2. Keeping the benchmark updated
>
> Thank you for pointing out the "time machine problem" -- the need to make benchmarks resistant to test set leakage into new models' training data.
>
> Overall, we are optimistic that the **ongoing nature** of the USA Computing Olympiad contests will help make the USACO benchmark leakage-resistant by allowing us to continuously update the benchmark with new contest problems.
>
> We (1) hope to release updated versions of the benchmark that include newer contest problems (2) have designed our benchmark to be easy to use in ways that both reveal and resist leakage, including standardized problem and solution formats that simplify supporting new and custom problems, as well as easy-to-use and efficient batch evaluation infrastructure that can reveal temporal effects by comparing performance on subsets of the benchmark data.
>
> As an example of the second point, in the paragraph of Section 3.1 in the paper entitled "Temporal effects are strong," we discuss an experiment that additionally evaluates on a subset of 36 problems released after training cutoffs and find that solve rates reduce, pointing to potential contamination in current models, even while they perform poorly on the vast majority of the benchmark.

---

> > ### Comment · Reviewer_hsvi · 2024-06-05
> >
> > Thanks for the response! Your strategy makes sense to me, I would strongly encourage maintaining an ever updating benchmark in the current world of LLMs!

---

### Decision · Program_Chairs · 2024-07-10

**Decision:**

Accept

**Comment:**

This paper introduces a new, very challenging benchmark based on USA Computing Olympiad contest questions. I am recommending acceptance based largely on the fact that this seem like it is a valuable new challenge task for LLMs.

I do have on lingering concern: reviewer hDmn raised a question about evaluation. The authors reported new experiments in their response, and they claim that the experiment doesn't fundamentally alter anything. However, this seems too simple. Their new results show a significant drop in performance once they introduce a train/test split. This could indicate that different splits would yield radically different result, or that the previous results were based on too much information from the test set. We just don't know. The qualitative results *were* impacted, though.

I think the above concern does not warrant rejecting the paper, but I would encourage the authors to think carefully about how to use their benchmark as a reliable tool for assessment and comparison.